# Family-Centered Advance Care Planning: What Matters Most for Parents of Children with Rare Diseases

**DOI:** 10.3390/children9030445

**Published:** 2022-03-21

**Authors:** Karen Fratantoni, Jessica Livingston, Sandra E. Schellinger, Samar M. Aoun, Maureen E. Lyon

**Affiliations:** 1Center for Translational Science, Children’s Research Institute, Children’s National Hospital, Washington, DC 20010, USA; kbfrat@gmail.com (K.F.); jessicalivingston95@gmail.com (J.L.); 2Division of General and Community Pediatrics, Children’s National Hospital, Washington, DC 20010, USA; 3Livio Health, 401 Harding St NE, Minneapolis, MN 55413, USA; sandra.schellinger@liviohealth.com; 4Perron Institute for Neurological and Translational Science, Nedlands, WA 6009, Australia; samar.aoun@perron.uwa.edu.au; 5Medical School, University of Western Australia, Crawley, WA 6009, Australia; 6School of Psychology and Public Health, La Trobe University, Melbourne, VIC 3086, Australia; 7Department of Pediatrics, George Washington University School of Medicine and Health Sciences, Washington, DC 20052, USA

**Keywords:** rare disease, advance care planning, decision-making, family caregiver, palliative care, psychosocial care, communication, pediatric

## Abstract

Few studies have described the goals and wishes of parents caring for their children with rare diseases, specifically when children are unable to communicate their preferences directly. The purpose of this study was to describe the parent’s understanding of their child’s illness, goals of care, and what mattered most to their child from the parent’s perspective. Six families completed a feasibility study of the FAmily CEntered (FACE)-Rare pACP intervention. Qualitative content analysis was performed on transcripts of videotaped responses to the Respecting Choices Next Steps pACP Conversation facilitated conversation guide about the goals of care. Codes were grouped into themes, with direct participant quotations representing the themes. Five themes emerged: getting out and moving freely; feeling included and engaged; managing symptoms and disease burden; coordinating care among many care team members; and managing today and planning for the future. In the context of pACP, families reported that what mattered most to their children included the freedom of movement and human connection and engagement, while parents strived to be effective caregivers and advocates for their child with a rare and severely disabling disease.

## 1. Introduction

Children with rare, life-limiting diseases face challenges to their physical and psychosocial health. A child diagnosed with a rare disease may require total life-long care, and a parent often provides that care for the duration of their child’s life. If a child cannot communicate, a parent must also try to determine their desires. There is no roadmap for this journey, leaving parents often feeling stressed or isolated [1]. Even though the parent is usually the most involved in the child’s care, there is a lack of literature on a parent’s experience caring for their child with a rare disease [2]. Few studies have described the goals and wishes of parents caring for their children with rare diseases, especially when children are unable to communicate their preferences directly.

There is a need for more research regarding parents’ experiences and desires when caring for children with rare diseases. In addition, specific information about their children’s medical conditions is often lacking, making it difficult to manage symptoms [3,4]. Often, little is known about future care needs [5,6,7]. General knowledge about the disease and its consequences is lacking [7], which may result in inadequate care [8]. There is also a gap in research focused on parents’ experience with the healthcare system [9], although recent studies focus on pathways to psychosocial care at a system level [10,11]. Overall, parents of children with rare diseases report that there is a lack of coordination of care [2]. In a survey administered toward 46 families of children with rare diseases in Australia, parents felt that the care of their children could be improved if there was better coordination of care and interaction between multiple providers [3]. 

Pediatric Advance Care Planning (pACP) is a process that facilitates discussions with parents and potentially with children about future medical treatment, care, and decisions if the child’s condition becomes life-threatening. It also helps families prepare for situations that may occur, and it can improve the family’s understanding of the child’s prognosis. Pediatric ACP is a core component of palliative care for children living with chronic conditions [12], and it should be offered to all parents of and children living with rare diseases.

The FACE-Rare intervention integrated two evidence-based processes: the Carer Support Needs Assessment Tool (CSNAT-Paediatric) Approach [13,14] and Respecting Choices^®^ Next Steps pACP conversation [15]. Quantitative outcomes of both aims were reported previously [16]. 

The aim of this article is to highlight the qualitative findings from the Respecting Choices^®^ Next Steps pACP session during which parents engaged in facilitated conversations about their understanding of their child’s illness, goals for their child, and what mattered to their children from parental perspectives. 

## 2. Methods

From October 2017 to January 2018, we recruited and enrolled parents of nonverbal children with rare diseases to participate in a feasibility study of a 4-session needs assessment and pACP intervention. Complete methodological details of the FACE-Rare intervention have been previously described [16]. Parents who were ≥18 years old, had a child with a rare disease, and spoke English were invited to participate. Participants’ children were between the ages of 1 and 24 years old, had a diagnosed rare disease, were not admitted to an intensive care unit during the study, were unable to participate in medical decision making due to age or disability, were not in foster care, and did not have an active Do Not Resuscitate Order. Children with the primary diagnoses of autism, rare cancers, HIV, sickle cell, cystic fibrosis, down syndrome, lupus, and muscular dystrophy were excluded, as previously published studies have investigated their disease-specific needs. Written informed consent was obtained from the parents, along with a signed waiver of assent for their child. The study was approved by the Institutional Review Board of Children’s National Hospital in Washington, DC, which is a quaternary hospital and the study site. 

### 2.1. Interviews

Table 1 illustrates the FACE-Rare study protocol timeline, which provides the context for the study results. The 4-session intervention was administered in person at the study site or by telemedicine. This study only reports on the qualitative data that emerged from Sessions 3 and 4, which consisted of structured goals of care conversation using the Respecting Choices Next Steps pACP conversation guide and a pACP document [15]. The structured conversation guide and pediatric Advance Care Plan are proprietary and are made available upon the completion of training and certification by Respecting Choices, a Division of C-TAC Innovations [15]. 

Respecting Choices is designed to be culturally sensitive to persons with disabilities [17]. This session was facilitated by two trainers from Respecting Choices, including one of the authors (SS). Interviews were videotaped, transcribed verbatim, and deidentified (using pseudonyms). A psychology graduate student/research volunteer transcribed and deidentified the videotapes. Transcriptions were then verified by the last author (ML). This study reports on the data that emerged from Sessions 3 and 4, which comprised responses to the Respecting Choices Next Steps pACP conversation guide and pACP care plan.

### 2.2. Data Analysis

The authors (K.F., J.L., and S.E.S.) performed conventional content analysis [18] on the transcribed Respecting Choices conversations using an iterative process. Four iterations occurred over a 4-month period from 4 March 2019 to 2 June 2019. For researcher characteristics, please refer to Biographies at the end of this article. 

J.L. and K.F. individually read the transcripts and assigned codes to transcripts. Codes were then jointly reviewed with S.E.S. and grouped into code families. Similar code families were combined into themes. Disagreements were resolved by consensus during monthly conference calls (K.F., J.L., S.E.S., and M.E.L.). Representative quotations illustrate key themes. Data were managed using hardcopy files. We used the Standards for Reporting Qualitative Research (SRQR) to guide our report [19]. 

## 3. Results

Eight parents were approached: One declined participation, and one was lost to follow-up after Session 1. Six parents completed Sessions 1, 2, 3, and 4. All participants were female, mean age 40 years (standard deviation = 7.7 years), 57% Caucasian, 29% black or African American, and 14% biracial. Two fathers participated with the mothers in Session 4 for the creation of an advanced care plan. The children ranged in age from 2 to 12 years with a mean age of 7 years. Five children had seizure related comorbidities, and all were technology dependent. Five used a wheelchair and feeding tube/pump, and one had a home ventilator. 

## 4. Key Themes

Five themes emerged throughout the structured interview, which not only followed specific questions. See Table 2, Table 3, Table 4, Table 5 and Table 6. The five themes about what matters most for parents of children with rare diseases were as follows: (1) getting out and moving freely; (2) feeling included and engaged; (3) managing symptoms and disease burden; (4) optimizing coordinated care among many team members; and (5) planning for the future. See Table 2, Table 3, Table 4, Table 5 and Table 6 for subthemes and illustrative quotations.

### 4.1. Theme 1: Getting out and Moving Freely

As illustrated in Table 2, parents emphasized the importance of encouraging independence for their children and providing opportunities to participate in activities. Physical exploration, optimizing mobility, and offering new experiences were all essential while simultaneously ensuring their safety. Creating and being a part of a supportive and inclusive environment were necessary to accomplish this goal, yet families were often reminded of the barriers to full inclusion. 

### 4.2. Theme 2: Feeling Included and Engaged

As illustrated in Table 3, parents want their children to feel connected to the outside world beyond their diagnosis and medical care. They desire meaningful interaction with others and opportunities to engage in enjoyable activities. 

### 4.3. Theme 3: Managing Symptoms and Disease Burden

As illustrated in Table 4, parents wanted to limit unnecessary interactions with the healthcare system, recognizing that their children with rare diseases have regular and frequent medical appointments, hospitalizations, and emergency department visits. Slowing the progression of disease and controlling symptoms are integral to a good quality of life. Many families mentioned the desire for seizure control, as seizures were disruptive to daily life and occupy attention that could be directed to other impactful therapies and interventions.

### 4.4. Theme 4: Optimizing Coordinated Care among Many Care Team Members

As illustrated in Table 5, parents wanted to optimize good care team communication means better care for children with rare diseases, especially at times of transition.

### 4.5. Theme 5: Managing Today and Planning for the Future

As illustrated in Table 6, parents expressed their hopes for the future: for their children to feel complete, whole, not a burden, thrive, and experience life to its fullest. Parents shared their fears about what the future may bring and their uncertainties about their ability to handle the challenges. 

## 5. Discussion

To the best of our knowledge, this is the first study to examine parents’ goals, values, and hopes for their child with a rare disease in the context of a structured family-centered pACP intervention. After completing the goals of care conversations with a trained/certified facilitator, all parents returned to complete an advanced care plan document [16]. Five themes emerged. With respect to living well, parents reported that their severely disabled children enjoyed getting out and moving freely, even though all but one of the children were wheelchair bound. Moreover, interactions with others so that they could feel included and engaged were also important to their children. Parents highlighted their caregiving role in managing symptoms and disease burden, coordinating care among many team members, and coping one day at a time by balancing their hopes and fears for their medically fragile child. Seizures were a common symptom associated with rare diseases [20], which caused significant distress. 

These findings are consistent with a scoping review that identified five themes with respect to the support needs of parent caregivers of children with a life-limiting illness [21]. The themes identified were support for communication; choice; information; practical information; and social, psychological, emotional, and physical needs. Unmet needs included support for siblings; respite care; out-of-hours care; and psychological, home, and educational support [21]. A focus group with adults living with a rare diagnosis also highlighted the value of participation in society [7].

Our family-centered approach, which included parents in the development phase of the protocol, is consistent with calls for a family-centered framework relative to pACP for children with medical complexities [22]. There is also agreement that pACP should be an ongoing process from the time of diagnosis for seriously ill children [23,24,25], which is especially important for children living with prognostic uncertainty. In our clinical experience, long-term relationships and trust develop between patients and families and their providers. As the medical condition progresses, a disconnect may develop between a parents’ beliefs, values, hopes, goals, and their child’s quality of life and developing a holistic treatment plan. The FACE-Rare intervention did not require the pACP facilitator to be well-known to the parents, as has been recently recommended [22,25]. Our findings are consistent with our previous trials of a three-session FACE pACP model, which demonstrated that the initial pACP conversations about the goals of care with parents of seriously ill children and the completion of an initial advance care plan can be successfully conducted by trained/certified facilitators that are not known to the parents, who are referred by their treatment team [16,26,27]. A summary of these conversations and the documents were then emailed by the facilitator to the treating clinician, and the facilitator places the documents in the medical record, laying the groundwork for future pACP conversations with their health care provider. Thus, FACE-Rare demonstrated families’ willingness to engage in goals of care conversations with a trained/certified nurse facilitator who was unknown to them. 

The FACE-Rare intervention is consistent with findings from focus groups with parents of children with medical complexities and their health care providers (HCPs) [22]. HCPs and parents expressed the desire that the patient and family be at the center of pACP discussions. HCPs noted the importance of taking time to recognize, understand, and support diversity and individuality between families. Parents also explained that the best pACP conversations were the ones in which they felt involved, respected, and accepted, which is similar to our findings. Parents identified topics that they felt should be included in pACP discussions, which were included in the FACE-Rare model: (1) quality of life, (2) beliefs and values, and (3) hopes and goals. In our study, the parents noted that their child’s quality of life was often underestimated by HCPs, thus highlighting the importance of asking parents about their child’s quality of life at baseline rather than making inferences based on their clinical status when admitted to the hospital, which is consistent with findings from a focus group with parents of children with medical complexities [22]. The focus groups with parents and HCP also thought a family’s values and belief system was foundational to pACP discussions, allowing HCP to better tailor care to each individual family [22], as was accomplished in the FACE-Rare intervention. Focus group parents and HCP also indicated that pACP discussions should include conversations surrounding their hopes and goals for their child because this process provided opportunities to collaboratively work toward and/or reframe hopes and goals [22]. Thus, study findings highlight the importance of incorporating parents’ hopes and values for what it means for their child to live well, prior to the completion of an advanced care plan. Understanding a parent’s focus on what is most important is possible with the FACE-Rare intervention. This approach may improve a parents’ ability to advocate on their child’s behalf and assist care team members to provide a person-centered care approach that matches the goals and values, with the needed care and resources as their child’s condition changes. 

## 6. Limitations

Research participants were selected to beta test the study protocol and to gain preliminary information on feasibility. Thus, we did not use sampling saturation as a criterion for our sampling strategy. This means that other themes might have emerged if we had continued sampling, which may limit generalizability. The cohort was small and from a single site. A larger pilot trial is ongoing to test the initial efficacy of FACE-Rare with 30 parents of children with ultra-rare diseases who are unable to participate in medical decision making [28]. 

The sex of the parent may have introduced bias. The primary caregiver in the home may depend on multiple factors, including parental employment; medical insurance; and the size, gender, and weight of the child. Clinically, many fathers have reported discomfort when caring for their female children with rare diseases once their child reach puberty age. 

Moreover, information that was not reported is also important. No parents reported service providers’ negative responses as a barrier for accessing services or assistance for their children, as has been reported in two small qualitative studies [5,6]. 

## 7. Conclusions

This study begins to close a gap in our knowledge [29] of parents’ goals of care for their children living with a serious illness who, in aggregate, constitute a significant proportion of pediatric inpatients with life-limiting conditions in tertiary and quaternary pediatric hospitals [30]. Children with rare diseases are part of a heterogeneous group and are often excluded from research [31], thereby creating a health disparities. Collecting qualitative data on patient and family member goals and wishes is a pivotal part of quality care. With information about the goals and wishes of the patients and families being discussed, accurate and appropriate recommendations for palliative and end-of-life care from the care team can be provided. Ongoing research will determine if the FACE-Rare pACP process of decision making for parents [27] adds benefits to clinical care and family well-being [32], as has been true with the FACE pACP model with adolescents with cancer and HIV and their families [26,27]. 

## Figures and Tables

**Table 1 children-09-00445-t001:** FACE-Rare Study Protocol Timeline.

Intervention	Session 1 Week 2	Session 2 Week 3	Session 3 Week 4	Session 4 Week 5
FACE-Rare	CSNAT	CSNAT	Respecting Choices Next Steps Conversation	Respecting Choices Next Steps pediatric Advance Care Plan

**Table 2 children-09-00445-t002:** Theme 1 Getting Out and Moving Freely.

Theme 1 Illustrative Quotations
*Supporting and Encouraging Independence* P1: “She has to be able to do some things that she wants to do… allow her to be as independent as possible” *Creating a supportive environment* P1: “…create an environment so that she can still be active as she can be or do what she would like to do.” *Optimizing mobility* P1: “Being able to be on the move. Just not being limited. When she’s in the activity chair, she’s in the feeding chair, she’s in the wheelchair, she’s not happy. She wants to explore. And so being able to move and being able to engage to the extent that she’s able to, is huge for her.” P1: “So, moving through a good day would be being active, exploring, not being limited and engaging…” *Ensuring safety during exploration* P2: “We come to the beach a lot because it’s, you know, peaceful and quiet. And it’s one-on-one attention with his grandparents when they come. And he gets the vacation experience in a safe way that we can control.” *Offering new experiences* P5: “Good quality of life is her being happy…exposing her to as much as we can, to give her the opportunity to experience … she can do a lot more than people expect.” P7: “He loves to get out of the house and go places. He really likes it if he’s in his wheelchair so that he can explore the new place. But not too loud.” *Disabilities can impede full inclusion* P5: “Hope…a world that is more inclusive. When leaving the house, because we do not have a child that is ambulatory, [it’s] hard to find a place that we can change a diaper because changing tables are only for infants and at 100 pounds, I can’t put her up on the changing table and yet if I put her on the floor, I can’t get her up from the floor. So, we’re limited between or changing her in the back seat of the car, and she’s an 11-year-old little girl, and so it limits her being included in a world the way we want her to be, and from us being in the world that we want.”

**Table 3 children-09-00445-t003:** Theme 2 Feeling Included and Engaged.

Theme 2 Illustrative Quotations
*Interact with others* P1: “She needs to be able to have some form of communication. She needs to have her sisters.” P5: “…being able to be with her peers but making accommodations to make it happen.” *Engaged in enjoyable activities* P1: “Being engaged. She loves her tablet; she’ll play her little computer games. She likes the swiping aspect. She can swipe and kind of navigate her way around.” P6: “Being held by someone and a massage”

**Table 4 children-09-00445-t004:** Theme 3 Managing Symptoms and Disease Burden.

Theme 3 Illustrative Quotations
*Limit unnecessary interactions with the healthcare system* P1: “Stay away from some of the orthopedic surgeries that she needs. I’m trying to hold off as long as I can.” P1: “Stays healthy… I need her to not have the cold or flu because things kind of snowball…keep her healthy and out of the healthcare system.” *Limit the progression of disease* P3: “Eliminate the chronic eye issue.” P7: “Worry about there being more issues…he had an ASD [Atrial Septal Defect] that was repaired a few years ago. But he started getting followed by a cardio myopathy clinic so, I get worried that something may turn up …like a new condition.” *Seizure control* P1: “So, my ultimate hope is much better seizure management…when we get better seizure management, then we’re better able to maximize speech therapy and receptive expression.” P3: “I’d love it if we’d reduce a lot of his seizure medication one day. Yeah, they sedate him…we’ve seen alertness change with medication, so it sure would be nice if we could reduce it someday.” *Optimize Quality of Life* P1: “…be as comfortable as possible. Her comfort would be of utmost concern” P7: “Address those baseline needs… I think he has a lot of little things where, very seemingly little things, that bother him and so he’s not comfortable. And that affects his ability to kind of grow and progress and learn.” *Minimize medications* P3: “Minimize how many medications he’s on and how we give them to him, so it maximizes his abilities and alertness.” *Self-Injuries* P7: “He’s banging his head on things. We’re worried about getting him too strong. You know, getting too big and too strong for me to be able to take care of everything… unintentional injuries, him injuring himself, something that will injure him.”

**Table 5 children-09-00445-t005:** Theme 4 Optimizing Coordinated Care among Many Team Members.

Theme 4 Illustrative Quotations
*Optimal communication and coordination among team members* P1: “That there’s better communication among our healthcare providers…nice if doctors got together once or twice a month to discuss … so they can get a good snapshot of where you are.” P6: “And this idea of, we’ll do the procedure, and we’ll work out the home care and the nursing after…that’s not going to work for us because there’s not going to be a place for us after. And those are the types of things that every thought, thoughts, you really have to forecast, way down the road. And that’s a lot of the concern, who’s going to do that? Who’s going to be willing to own that and am I going to get team members that are really going to work with us?” *Preparing for transition* P3: “I guess looking ahead with [his] age, parents like us have a discussion with the transition to adult care so I brought that up to his physicians periodically, and they assured me, don’t worry about him, especially with a rare disease, you know. He’s, he’s not going to be sent away and he’s at a children’s hospital, I mean I know but he’s not your typical patient that can go anywhere. A huge concern for when we are away and we’ve had to seek medical care, we seek help, other hospitals are just not equipped for him and we’ve had doctors tell us, we’re just not equipped to helping [him]. And so, that just makes your heart sink and you always want him to be in the hands of someone who knows him best and have already done such good already in managing him.”

**Table 6 children-09-00445-t006:** Theme 5 Planning for the Future.

Theme 5 Illustrative Quotations
*Hopes for the future* P2: “So, I want him to thrive as much as he can, for as long as he can, but I don’t want him to hang on just for me or for my husband or parents. I want his life to be his life and that for him, to leave hopefully or a little more peacefully.” P6: “I hope that we’re able to try to be able to give her the best possible life possible and that she knows that we do our best to make that happen and that she feels loved and she feels acknowledged as an individual and that she knows, despite all of it, that we think she is a wonderful, miraculous little girl. And that she lives the best life and that we’re able to support that.” P7: “…stability and being able to be happy and enjoy each other… not stressed out worrying about who will take care of him or working or money or things like that.” *Fears* P5: “If that day comes …. I kind of like want to be there and to hold her hand and to let her know it’s okay. I don’t want [her] to go through that alone and I think for me, that’s my biggest fear, is her having to go through that alone and suffer.” P7: “When I’m not here anymore, who’s going to take care of him?” *Uncertainty of what might happen next-that the family may not be able to handle* P2: “Every ED visit is more like-fear oriented…I fear every hospitalization is one step closer to something horrible and disease progressing out of what we can manage right now.” P3: “… we have everything so well managed and so there’s always there’s that little concern with, what if a new challenge arises and the current method of treatment doesn’t meet the needs, can’t take care of the challenges. So, whole team is very proactive with keeping that under control. Just you know, his medical needs never outweigh the available treatment.”

## Data Availability

Only transcribed data can be obtained, as the original source videotapes contain personal information. Contact the last author at mlyon@childrensnational.org for additional information.

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
