# Peer review of "Family-Centered Advance Care Planning: What Matters Most for Parents of Children with Rare Diseases"

_children, 2022, doi:10.3390/children9030445_

Round 1

Reviewer 1 Report

I find your manuscript novel and interesting, although I have a few minor questions and suggestions to improve readability and clarity of your work.

In Abstract

I wonder about the word “during” on page 16, is it only during the pACP, you are interested in parents hopes and meanings…. Only “during” pACP??

I find it confusing with FACE-Rare pACP and the Respecting Choices Next Steps pACP, what is part of this report? Please clarify and maybe leave out FACE-Rare here??

It would be helpful to know that five themes emerged.

The aim of the study it is not very clear.

In abstract it is stated “The purpose of this study was to describe parental hopes and the meaning of living well for their children with rare diseases during pediatric advance care planning (pACP).”

In Introduction it is stated “The aim of this article is to highlight the qualitative findings from the Respecting Choices® Next Steps pACP session during which parents engaged in facilitated conversations about their understanding of their child’s illness, goals for their child, and what mattered the most to them.” Who is them referring to, parents or child?

Method

I wonder is it necessary to inform the reader about the FACE-Rare intervention? Page 2 line 70-71

On line 86 another instrument is mentioned CSNAT Paediatric, it is much confusing. I would like to see a scheme / figure of study design and instruments that is part of the report. I assumed from the reading that this report comes from data generated by CSNAT Paediatric session 3 & 4 and Respecting Choices Next Steps PACP conversation guide and a pACP document. Although I am not sure. In line 94 and 95 it is stated “his study reports on the responses to the Respecting Choices Next Steps pACP conversation guide”. Yet, from that sentence it is not clear if the data emerge from the CSNAT-Paediatric sessions?? Please clarify.

Results

There is an error on line 110, All participants were female, mean age 40 years (SD?), 57% Caucasian, 29% black or African American, and 14% biracial. Two fathers participated with the mothers in Session 4 for creation of the advance care plan.

Discussion

It strikes me that the authors are convinced this is the first study, it might be the first published study. It would give a humbler impression if the added “To our knowledge this is the first……. “

It would be good to know if the themes are following the questions in the Respecting Choices Next Steps conversation guide and the pACP document, or not?

It is unclear with the correlation of this report and the FACES study often referred to; this needs to be clarified. 

It may be due to me not being English in native tongue, but I would suggest the authors to rephrase the sentence on line 222.

Author Response

Reviewer 1

I find your manuscript novel and interesting, although I have a few minor questions and suggestions to improve readability and clarity of your work.

 In Abstract

I wonder about the word “during” on page 16, is it only during the pACP, you are interested in parents hopes and meanings…. Only “during” pACP?

-Thank you for this feedback, we have removed the word “during” on page 16. You are correct it is not only during pACP that we are interested in parents hopes and meanings.

I find it confusing with FACE-Rare pACP and the Respecting Choices Next Steps pACP, what is part of this report? Please clarify and maybe leave out FACE-Rare here??

-We have rewritten this section to clarify that it is the Respecting Choices Next Steps pACP that is part of this report. We have decided to keep the reference to the FACE-Rare protocol to give the reader the context for what occurred prior to the Respecting Choices interview.

It would be helpful to know that five themes emerged.

-We have added the phrase “five themes emerged” to the abstract and the text.

The aim of the study it is not very clear.

In abstract it is stated “The purpose of this study was to describe parental hopes and the meaning of living well for their children with rare diseases during pediatric advance care planning (pACP).”

In Introduction it is stated “The aim of this article is to highlight the qualitative findings from the Respecting Choices® Next Steps pACP session during which parents engaged in facilitated conversations about their understanding of their child’s illness, goals for their child, and what mattered the most to them.” Who is them referring to, parents or child?

-We have rewritten to consistently describe the aim of this article in the abstract and introduction.

-We have clarified the sentence to read “and what mattered the most to their child from the parent(s) perspective.”

Method

I wonder is it necessary to inform the reader about the FACE-Rare intervention? Page 2 line 70-71?

- We have decided to keep the reference to the FACE-Rare protocol to give the reader the context for what occurred prior to the Respecting Choices interview.

On line 86 another instrument is mentioned CSNAT Paediatric, it is much confusing. I would like to see a scheme / figure of study design and instruments that is part of the report. I assumed from the reading that this report comes from data generated by CSNAT Paediatric session 3 & 4 and Respecting Choices Next Steps PACP conversation guide and a pACP document. Although I am not sure. In line 94 and 95 it is stated “his study reports on the responses to the Respecting Choices Next Steps pACP conversation guide”. Yet, from that sentence it is not clear if the data emerge from the CSNAT-Paediatric sessions?? Please clarify.

-We thank you for bringing to our attention the lack of clarity here. We have added a table for the study design and clarified this study is only about the qualitative analysis of the transcripts from the Respecting Choices conversations in Sessions 3 and 4.

- We have added the following sentence with respect to the instrument/structured interview guide and pACP advance care plan: “The structured conversation guide and pACP guide are proprietary and made available upon completion of training and certification by Respecting Choices, a Division of C-TAC Innovations at https://respectingchoices.org/”

Results

There is an error on line 110, All participants were female, mean age 40 years (SD?), 57% Caucasian, 29% black or African American, and 14% biracial. Two fathers participated with the mothers in Session 4 for creation of the advance care plan.

-We have provided the SD of age for participants, 7.7 years. Thank you for catching this error.

Discussion

It strikes me that the authors are convinced this is the first study, it might be the first published study. It would give a humbler impression if the added “To our knowledge this is the first……. “

-Thank you for this guidance. We have changed the wording per your recommendation to “To our knowledge, this is the first…”

It would be good to know if the themes are following the questions in the Respecting Choices Next Steps conversation guide and the pACP document, or not?

-Thank you for raising this very important question. We have added this to the results section,” Five themes emerged throughout the structured interview not just following specific questions, as we found that the themes occurred spontaneously throughout the conversation and were coded as they occurred.”

Reviewer 2 Report

Review on Family Centered Advance Care Planning: What matters most for parents of children with rare diseases

The authors present a feasibility study to highlight qualitative findings of an assessment and intervention for families of children with rare diseases. The study is designed to identify the parental understanding of their child’s illness, goals for their child, and what matters most to them. Data of six families of nonverbal children with rare diseases were included. The authors identified freedom of movement and human connection and engagement for their children, as well as striving to effective caregivers and advocates for their child as most important for parents. This is an interesting paper with relevance to researchers and clinicians working in this field. The qualitative study is well presented and a good fit for the journal. I have, however, some comments to do:

Minor comments and revisions

  1. In the introduction section (p.1, line 43-45), there is more recent literature on family’s experience with healthcare systems (e.g., Witt et al., 2021).
  2. In the results section (p.3, line 110) a question mark is still given behind the SD. A corresponding result should be inserted.
  3. It should be discussed whether the gender of the parents may have biased the results.

Author Response

Reviewer 2

The authors present a feasibility study to highlight qualitative findings of an assessment and intervention for families of children with rare diseases. The study is designed to identify the parental understanding of their child’s illness, goals for their child, and what matters most to them. Data of six families of nonverbal children with rare diseases were included. The authors identified freedom of movement and human connection and engagement for their children, as well as striving to effective caregivers and advocates for their child as most important for parents. This is an interesting paper with relevance to researchers and clinicians working in this field. The qualitative study is well presented and a good fit for the journal. I have, however, some comments to do:

 Minor comments and revisions

  1. In the introduction section (p.1, line 43-45), there is more recent literature on family’s experience with healthcare systems (e.g., Witt et al., 2021).

-Thank you for the reference to the more recent publication on the family’s experience with the healthcare systems. We have included this citation and an additional reference to an ongoing trial of a family-based intervention for children and families affected by rare disease in Germany. See citations 10 and 11.

  1. In the results section (p.3, line 110) a question mark is still given behind the SD. A corresponding result should be inserted.

-What is the SD? 7.7 years. We have corrected this error.

  1. It should be discussed whether the gender of the parents may have biased the results.

-We have added to the limitations of this study: “We do not know how the gender of the parents may have biased the results. Who is the primary caregiver in the home may depend on medical insurance coverage in the United States (mother’s job provides it and father’s does not), the size, gender, and weight of the child. For example, a news story reported a father was the primary caregiver of their girl with a rare disease, but he felt it was not appropriate for him to continue to care for her once she reached puberty. So, this is a very important and complicated question”.